

# Community shelter use in response to two benthic decapod predators in the Long Island Sound

David M. Hudson[1,2], Dugan Reagan[1] and Joseph F. Crivello[1]

[1] Department of Physiology and Neurobiology, University of Connecticut, Storrs, Connecticut, United States
[2] Division of Science, Mathematics, and Health Professions, Atlanta Metropolitan State College, Atlanta, Georgia, United States

## ABSTRACT

To investigate community shelter effects of two invasive decapod species, *Hemigrapsus sanguineus* and *Carcinus maenas*, in the Long Island Sound (LIS), we deployed artificial shelters in the intertidal and immediate subtidal zones. These consisted of five groups during the summer: a control, a resident *H. sanguineus* male or female group, and a resident *C. maenas* male or female group. We quantified utilization of the shelters at 24 h by counting crabs and fish present. We found significant avoidance of *H. sanguineus* in the field by benthic hermit crabs (*Pagurus* spp.) and significant avoidance of *C. maenas* by the seaboard goby (*Gobiosoma ginsburgi*). The grubby (*Myoxocephalus aenaeus*) avoided neither treatment, probably since it tends to be a predator of invertebrates. *H. sanguineus* avoided *C. maenas* treatments, whereas *C. maenas* did not avoid any treatment. Seasonal deployments in the subtidal indicated cohabitation of a number of benthic species in the LIS, with peak shelter use corresponding with increased predation and likely reproductive activity in spring and summer for green crabs (*C. maenas*), hermit crabs (*Pagurus* spp.), seaboard gobies (*G. ginsburgi*), and grubbies (*Myoxocephalus aenaeus*).

## INTRODUCTION

In marine systems, the availability of shelter and risk avoidance when choosing shelter has a profound effect for many animals on growth rates, survival, avoidance of predators, and particularly mating systems for decapod crustaceans (*Atema, 1986*; *Perkins-Visser, Wolcott & Wolcott, 1996*; *Beck, 1997*; *Wieters et al., 2009*). As shelter is often at a premium (*O'Neill & Cobb, 1979*), conferral of group defense can be beneficial. For example, juvenile spiny lobsters in the Florida Key exhibit cohabitation within and among species within the same shelter (*Childress & Herrnkind, 2001a*; *Childress & Herrnkind, 2001b*; *Jordan, 2010*). As this can result in negative competitive interactions, this requires the animal to evaluate the risk associated with cohabitation (*Jordan, 2010*). Many marine species

Corresponding author
David M. Hudson,
dhudson@atlm.edu

have to select new shelter when they migrate into, out of, and within estuaries seasonally or with changes in abiotic factors such as dissolved $O_2$ or salinity (*Pörtner, 2001*; *Ortiz-León, de Jesús-Navarrete & Cordero, 2007*). Shelter use by different benthic species varies by season, since the specific shelter challenges faced by resident species with the presence of migratory species or influence of non-native species can vary by time of year and competition between animals (*Perkins-Visser, Wolcott & Wolcott, 1996*; *Childress & Herrnkind, 2001a*). In this study, we show Long Island Sound (LIS) benthic species not only modify their shelter use by season, but also by the presence of invasive decapod predators or competitors.

Decapod crustacean populations can be limited by sufficient access to shelter (*Perkins-Visser, Wolcott & Wolcott, 1996*; *Childress & Herrnkind, 2001a*). Particularly, decapod crustaceans, especially crabs and lobsters, are known to utilize benthic habitat differently depending on the sex of the individual (*Hines, Lipcius & Haddon, 1987*; *Karnofsky, Atema & Elgin, 1989*), and can behaviorally respond to the presence of only one individual in a shelter (*Cowan & Atema, 1990*). Shelter's importance is also evident since mortality in the losing group can increase when interspecific and intraspecific competition in marine systems for quality shelter limits its accessibility (*O'Neill & Cobb, 1979*; *Cobb, 1981*; *Richards & Cobb, 1986*; *Grove & Woodin, 1996*; *Brousseau, Kriksciun & Baglivo, 2003*; *Jordan, 2010*). In the LIS intertidal and subtidal zones, both the European green crab, *Carcinus maenas* (Linnaeus, 1758), and the Asian shore crab, *Hemigrapsus sanguineus* (De Haan, 1835) are invasive species. These species are competitors for space and food with native species (*MacDonald et al., 2007*; *Griffen, Guy & Buck, 2008*), requiring the identification of those finfish and crustacean species that may also utilize the same shelter and are likely to be affected in currently-invaded areas in order to predict likely future effects on community members.

The invasion of *H. sanguineus* to the western North Atlantic Ocean supplanted the earlier invader, *C. maenas*, and *H. sanguineus* is currently invading Europe along with its congener, *Hemigrapsus takanoi* (*Lohrer & Whitlatch, 2002*; *Asakura & Watanabe, 2005*; *van den Brink, Wijnhoven & McLay, 2012*; *Gothland et al., 2013*; *Gothland et al., 2014*). The Asian shore crab, *H. sanguineus*, was first found on the U.S. east coast in New Jersey in 1988 (*Williams & McDermott, 1990*). Since then, it became the most dominant crab in the intertidal zone of the northwest Atlantic Ocean, with densities up to 300 individuals/$m^2$ (*McDermott, 1998*; *Lohrer & Whitlatch, 2002*; *Kraemer et al., 2007*). It is found subtidally, but in lower densities (D. Hudson, 2011, personal observations). *H. sanguineus* is a generalist omnivore and can survive in a wide range of salinities (*Depledge, 1984*; *Hudson, 2011*). It prefers rocky substrate to sand and settles in the presence of conspecific adult olfactory cues (*Lohrer et al., 2000*; *Steinberg, Epifanio & Andon, 2007*; *Hudson, 2011*; *Rasch & O'Connor, 2012*). In the intertidal zone, predation occurs from both terrestrial and marine sources, which makes *H. sanguineus* a vigorous intertidal competitor for space and shelter due to its avoidance of risk during shelter choice and aggression toward smaller interspecific competitors (*Jones & Shulman, 2008*; *Wieters et al., 2009*; *Rasch & O'Connor, 2012*; *Peterson et al., 2014*). As such, we wanted to see if the presence of this species impacted the use of shelter by LIS community members in the field.

The previous intertidal resident invader, the European green crab, *C. maenas*, is still seen in the intertidal and subtidal zone in the LIS, though in lower densities than in its home range (*Amaral et al., 2009*). The arrival of *C. maenas* occurred sometime pre-1817 (*Say, 1817*) and that of *H. sanguineus* in the late 1980s (*Williams & McDermott, 1990*). However, *C. maenas* itself is still a potent invasive species worldwide and on the North American continent (*Freeman & Byers, 2006*; *Darling et al., 2008*). As benthic animal survival is so tied to the availability and ability to maintain shelter (*O'Neill & Cobb, 1979*; *Cobb, 1981*; *Richards & Cobb, 1986*; *Grove & Woodin, 1996*; *Brousseau, Kriksciun & Baglivo, 2003*; *Jordan, 2010*), our main focus for this work was to elucidate the sublethal choice effect that these two species have on the use of shelter by native fauna in the LIS. This information can be used in areas where both of these species are still relatively new invaders, to allow for an informed management strategy of native species.

Our main objectives for this study were to: (1) capture any sexual variability in shelter use by decapods, as described above for other decapods, which was accomplished through a field experiment with empty PVC artificial shelters as a control, and a male and female treatment for each of the two types of crab (*H. sanguineus* and *C. maenas*), for a total of five treatments; and (2) use these deployments and complementary laboratory studies (to better explain any effects of *C. maenas* and *H. sanguineus* on one another) to predict which LIS community species found in our shelters will be most affected by the presence of one of these invaders.

## MATERIALS AND METHODS

### Community shelter use

To measure LIS community shelter use differences in the field, we deployed a field shelter behavior assay. Benthic animals in the LIS are dependent on shelter for survival, so these shelter tubes imitate shelters that are unoccupied or occupied to test the potential for cohabitation with two important intertidal invaders in a shelter-limited area adjacent to habitat with ample shelter (small boulders). Replicates of the field apparatus (Fig. 1B) were deployed during the summer of 2009 for approximately 24 h in the immediate subtidal zone (~1 m depth) and intertidal zone (0.25 m above Mean Lower Low Water (MLLW), the lowest low tide daily) of the northwest end of Pine Island, Groton, CT, USA (41°18′47.434″N, 72°3′36.216″W). The tubes were at least 1.5 m apart, and were not attached to one another, but were all deployed along a uniform depth parallel to a shoreline with similar bottom characteristics along the length of the deployment. In cursory diving surveys, mobile animals seemed varied by species and fairly evenly distributed along the deployment location. This should have avoided the initial species approach phenomenon as much as possible. We operated under Connecticut Department of Environmental Protection Scientific Collector's Permits # SC-06040 and # SC-09015. Other researchers have quantified decapods' responses artificial shelter with conspecific presence (*Nevitt et al., 2000*; *Childress & Herrnkind, 2001a*), so the apparatus was designed to fit the target species. The apparatuses were constructed with weathered (3+ years exposed to the elements) 5.08 cm PVC tubing, with two 30 cm sections connected by a PVC joint. Cages constructed of 0.5 cm plastic mesh were used to isolate the live treatment

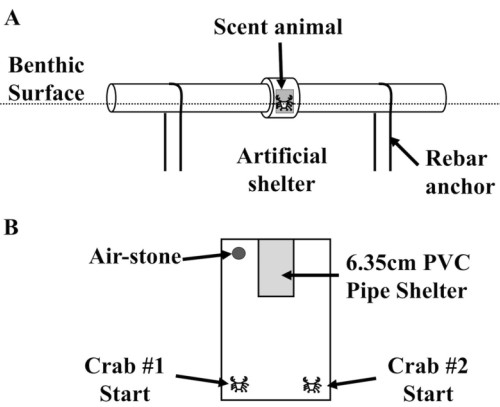

**Figure 1 Field and laboratory shelter setup.** Over the course of several weeks, the diagrammed an artificial shelter was deployed at the northwest side of Pine Island, USA (41°18′47.434″N, 72°3′36.216″W) by fixing it to the sediment for a 24 h soak of the treatments described in the Methods. (B) Shelter setup for direct shelter competition experiments between *Hemigrapsus sanguineus* and *Carcinus maenas*, which was completed in an 18 × 31 cm plastic box with sand substrate for 24 h. A 10 × 5.0 cm shelter, made from 6.35 cm PVC pipe with a quarter of its circumference removed, was arranged in the container to have only one useable opening.

animal inside the tube. We saw some influence of individual presence on one another with different species in the laboratory, and the original idea was to pump (expensively) water over an individual in the field and into the tubes. We did not have sufficient funding for that design, so we were confident in using a single "bait" animal. Control tubes were deployed with an empty cage to provide internal structure, and all tubes were anchored with U-shaped steel rebar. After a 24 h soak, divers capped each end, and deployed the next apparatus. The tubes were deployed over a 2-week period with random assignment of treatments each day in groups of 20–25 tubes, until 60 replicates were reached for each treatment. Capped tubes were quantified each day for the presence of crab and arthropod species, and other macroinvertebrates and vertebrates. Catch Per Unit Effort (CPUE) is reported here as percent of shelters used (%) calculated as the number of animals caught in each artificial shelter treatment, divided by the number of shelters for that treatment (60 per treatment). Data was analyzed in Systat 12.0 via one-way ANOVA, and Tukey's post-hoc test ($\alpha = 0.05$).

We took two approaches: (1) to capture animals that may only be using subtidal shelters seasonally, we deployed the artificial shelters four times over the course of a year; (2) In order to gain a sense for the intertidal use of shelter, which is most used in the summer months in the LIS, we also deployed intertidal experimental artificial shelters during the summer.

## Seasonal subtidal deployment

Due to dominance of *Hemigrapsus sanguineus* in the intertidal system, we wanted to see the overall effect on community shelter use across four seasons for this species in the subtidal zone. These shelters were deployed subtidally (~1 m depth) across four seasons in 2009, with control, and live male and female *H. sanguineus* bait treatments (total N = 180 for each deployment).

## Summer subtidal and intertidal deployment

In the summer months two additional treatments; live male and live female *Carcinus maenas* bait treatments were added. Thus, five treatments were used, along with the same control, live male and live female *H. sanguineus* treatments as already deployed in the seasonal deployment. We deployed the artificial shelters both in the subtidal and in the intertidal zone. As such, for summer experiments, each treatment was deployed in replicates of 60 both subtidally and intertidally (total N = 300 intertidal, N = 300 subtidal). A two-way ANOVA with tidal deployment and treatment was used to determine the interaction between treatment and tidal deployment, and size differences between subtidal and intertidal deployments were analyzed with Student's t-test.

## Shelter interaction for *H. sanguineus* and *C. maenas*

To further determine size effects on dominance and shelter use of the two invaders, shelter competition experiments utilized size ratio for interspecific and intraspecific shelter dominance between *H. sanguineus* and *C. maenas*. Multiple carapace width ratios, approximately 3:1, 2:1, 1:1, 1:2, and 1:3 in replicates of 10, were used for a one-on-one laboratory shelter competition experiment that lasted 24 h. We compared intraspecific (*H. sanguineus* or *C. maenas* versus member of the same species) and interspecific (*H. sanguineus* versus *C. maenas*) competition trials. Shelter use frequency differences between competitors were analyzed using Fisher's Exact Test. The effect of size ratio of competitors on "winner" and mortality was analyzed by a linear regression in R statistical software. Crabs were introduced to a plastic 18 × 31 cm arena with 2.5 cm of sand substrate and a 10 × 5.0 cm shelter made from 6.35 cm PVC pipe with a quarter of its circumference removed arranged in the container to have only one useable opening (Fig. 1B). A clear piece of plexiglass was placed over the container to prohibit escape, and an airstone oxygenated the tank. Each competition experiment involved placing two individuals on opposite sides of the container, width-wise, while the shelter was opposite both of the individuals, length-wise. The two individuals in the container were then allowed to compete for the shelter over a 24 h period, after which the positions of the crabs were noted as in the shelter or outside the shelter. Injuries to the claws or legs of each crab due to fighting were recorded. The plastic container, PVC pipe, and sand were cleaned with 10% bleach between replicates.

# RESULTS

## Community shelter use

Four species commonly cohabitated in the subtidal artificial shelters: the green crab *Carcinus maenas*, hermit crabs *Pagurus* spp. (grouped by genus), the seaboard goby *Gobiosoma ginsburgi*, and the grubby *Myoxocephalus aenaeus*. Subtidal deployment caught very few *Hemigrapsus sanguineus* individuals (11 individuals in 960 replicates deployed). Seasonal change in shelter use (with no crabs present, or with *H. sanguineus* male, and *H. sanguineus* female present) occurred amongst the four most collected species (Fig. 2).

No significant difference (one-way ANOVA, F = 1.394, 4 d.f., p = 0.236, N = 300) was seen for *C. maenas* subtidal shelter preference by treatment. There is no indication of

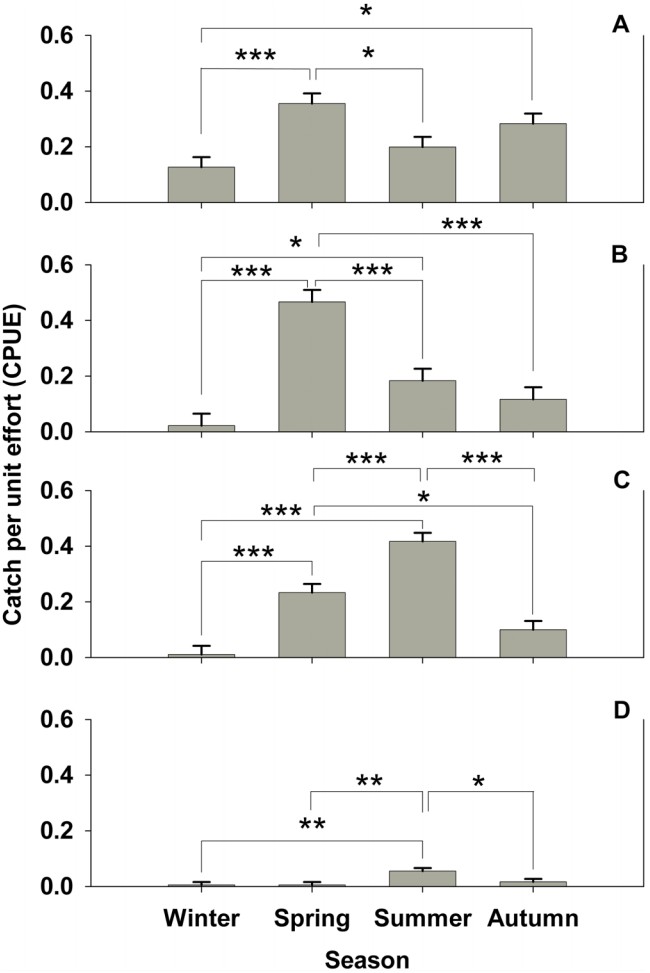

**Figure 2 Seasonal subtidal shelter use.** Seasonal subtidal shelter use across seasons for all treatments for which there are seasonal data (control, *Hemigrapsus sanguineus* male, and *H. sanguineus* female) in catch per unit effort (CPUE, in number of animals per number of tubes (n = 180 tubes for each bar)) mean ± SE for: (A) *Carcinus maenas*, (B) *Pagurus* spp., (C) *Gobiosoma ginsburgi*, and (D) *Myoxocephalus aenaeus*. Significance between seasons indicated by brackets: "*" = p < 0.05, "**" = p < 0.01, "***" = p < 0.001.

habitat partitioning by ontogeny for *C. maenas*, since carapace widths for summer subtidal and intertidal areas were not significantly different (mean = 25.4 mm for intertidal, mean = 25.64 mm for subtidal, Student's t-test, p = 0.95). However, significant seasonal shelter usage changes were observed in *C. maenas* (one-way ANOVA, F = 7.38, 3 d.f., p < 0.001, N = 721). Spring shelter usage was nearly triple that of winter (Tukey's post-hoc test, α = 0.05, p < 0.001), which then significantly (Tukey's post-hoc test, α = 0.05, p < 0.05) dropped by 54% in the summer deployment (Fig. 2A). *C. maenas* also doubled its shelter use in autumn over the previous winter (Tukey's post-hoc test, α = 0.05, p < 0.05 over winter values) (Fig. 2A). When all four seasons' subtidal data are pooled, in which there were control, *H. sanguineus* male and *H. sanguineus* female treatments, *C. maenas* showed a trend for avoidance of the male *H. sanguineus* treatment over the control (one-way ANOVA, F = 2.71, 2 d.f., p = 0.067, N = 721, Tukey post-hoc

p = 0.068, p = 0.200, for *H. sanguineus* male and *H. sanguineus* female treatments, respectively).

Other subtidal organisms showed changes in seasonal shelter use in our data. Pagurids changed their use of shelter (one-way ANOVA, F = 19.399, 3 d.f., p < 0.001, N = 721; all pairwise comparisons by Tukey's post-hoc test, α = 0.05, p < 0.001), with spring shelter usage 23 times that of winter, then a subsequent significant dip during the summer by about one third, and further dip from spring values during autumn to about 11 percent shelter use (Fig. 2B). The seaboard goby *Gobiosoma ginsburgi* (Fig. 2C) showed a significant seasonal signal as well (one-way ANOVA, F = 32.425, 3 d.f., p < 0.001, N = 721; all pairwise comparisions by Tukey's post-hoc test, α = 0.05, p < 0.001), with a significant increase in shelter use by about 23 times from winter to spring, and a doubling of spring values during summer. Similarly, *Myoxocephalus aenaeus* (Fig. 2D) significantly (one-way ANOVA, F = 5.061, 3 d.f., p = 0.002, N = 721) increased its shelter use during summer versus winter and spring (Tukey's post-hoc test, α = 0.05, p < 0.005) and decreased its shelter use in autumn (Tukey's post-hoc test, α = 0.05, p < 0.05).

Organisms were only found in sufficient numbers to merit comparison by treatment in summer. As for subtidal summer shelter use, *Pagurus* spp. showed preference for treatments (one-way ANOVA, F = 4.017, 4 d.f., p = 0.003, N = 300), specifically avoiding both *H. sanguineus* male and female treatments (Tukey's post-hoc test, α = 0.05, p < 0.03) in the subtidal (Fig. 3A) and using these shelters one to two times less often than others. The goby *G. ginsburgi* also showed preference for specific treatment shelters (one-way ANOVA, F = 15.277, 4 d.f., p < 0.001, N = 300) by avoiding completely (Tukey's post-hoc test, α = 0.05, p < 0.001) both of the *C. maenas* treatments (Fig. 3B). The grubby *M. aenaeus* did not show preference (one-way ANOVA, F = 0.823, 4 d.f., p = 0.511, N = 300) for any treatments, and was found at lower catch rates (data are in attached Supplemental Data).

Only *C. maenas* and *H. sanguineus* were found in sufficient numbers in summer intertidal deployments to merit analysis by treatment. In those, *C. maenas* showed a significantly lower intertidal shelter use (one-way ANOVA, F = 29.290, p < 0.001, N = 600) than subtidal shelter use by a factor of about 3.6 times (Fig. S1A). For summer deployments, no significant difference nor trend (one-way ANOVA, F = 1.394, p = 0.236, N = 300) was observed for subtidal shelter preference by treatment in *C. maenas* (Figs. S1B and S1C). *H. sanguineus* exhibited the opposite pattern, with significantly lower (one-way ANOVA, F = 30.88, 1 d.f., p < 0.001, N = 600) subtidal shelter use (1% use at 3 crabs in 300 shelters) than intertidal shelter use (14% use at 42 crabs in 300 shelters) (Fig. S2). No comparison by treatment was possible for *H. sanguineus* in subtidal treatments due to low catch rate, but catch efficiency was approximately 25% overall in the intertidal zone.

In the summer intertidal experiments, differences in male and female *H. sanguineus* behavior were observed. *H. sanguineus* males used shelters of any treatment 8.3% of the time (N = 301 shelters), but used intertidal treatments differently (one-way ANOVA, F = 3.46, 4 d.f., p = 0.0088, N = 301). Males displayed a lower (3.3% of treatment shelters) use of *C. maenas* male (Tukey post-hoc, p = 0.029) and *C. maenas* female (1.6% of total shelters) treatments (Tukey post-hoc, p = 0.011) when compared to the 16.9% use of

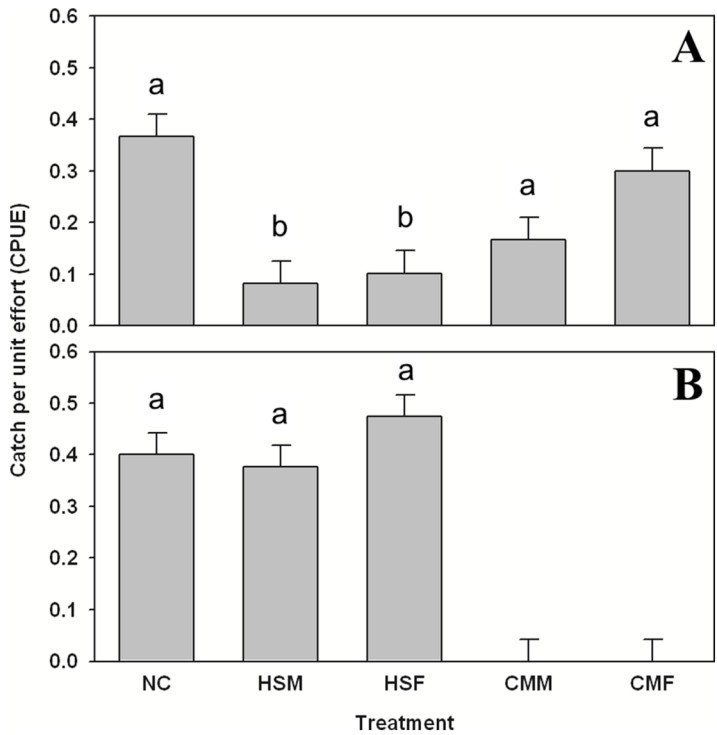

**Figure 3 Summer subtidal shelter use in hermit crabs and gobies.** Summer subtidal shelter use across treatments in catch per unit effort (CPUE, in number of animals per number of traps (n = 60 for each bar)) mean ± SE for (A) hermit crab *Pagurus* spp. and (B) seaboard goby *Gobiosoma ginsburgi.* Treatments: NC = No Crab, HSM = *H. sanguineus* Male, HSF = *H. sanguineus* Female, CMM = *C. maenas* Male, and CMF = *C. maenas* Female. Bars with the same letter are statistically similar (p > 0.05).

*H. sanguineus* female treatment shelters (Fig. 4A). Female *H. sanguineus* crabs also differed significantly in use of intertidal shelters (one-way ANOVA, F = 2.577, 4 d.f., p = 0.038, N = 301), but across all treatments only used 4.65% of them (N = 301 shelters). Female *H. sanguineus* crabs (Fig. 4B) did not choose conspecific *H. sanguineus* male treatment shelters (none were used, 0%) when compared with the control 11.7% usage rate (Tukey post-hoc, p = 0.018). There were no significant differences for other treatments, perhaps due to females' low shelter usage rate.

When pooled regardless of sex or maturity, significant differences were found in use of intertidal treatments by *H. sanguineus* (one-way ANOVA, F = 5.39, 4 d.f., p = 0.0003, N = 306). In intertidal treatments, *H. sanguineus* showed no significant reduction in shelter use of either *H. sanguineus* treatments over control (cage only) treatment (Tukey post-hoc, p > 0.05), sticking at 25% (15/60) use for control shelters, 8.33% (5/60) use for *H. sanguineus* male treatments, and 25.4% (15/59) use for *H. sanguineus* female treatments. The greater usage of *H. sanguineus* female shelters over all other crab treatment shelters is significant (*H. sanguineus* male, and both *C. maenas* male and female treatments, Tukey post-hoc, p = 0.040, p = 0.006, p = 0.002, respectively). However, *H. sanguineus* did significantly decrease use of *C. maenas* female treatment shelters to 4.84% (3/62) (Tukey post-hoc, p = 0.031), and trended (Tukey post-hoc, p = 0.064) to decrease (6.67%, or 4/60) use of *C. maenas* male treatment shelters (Fig. 4C). When

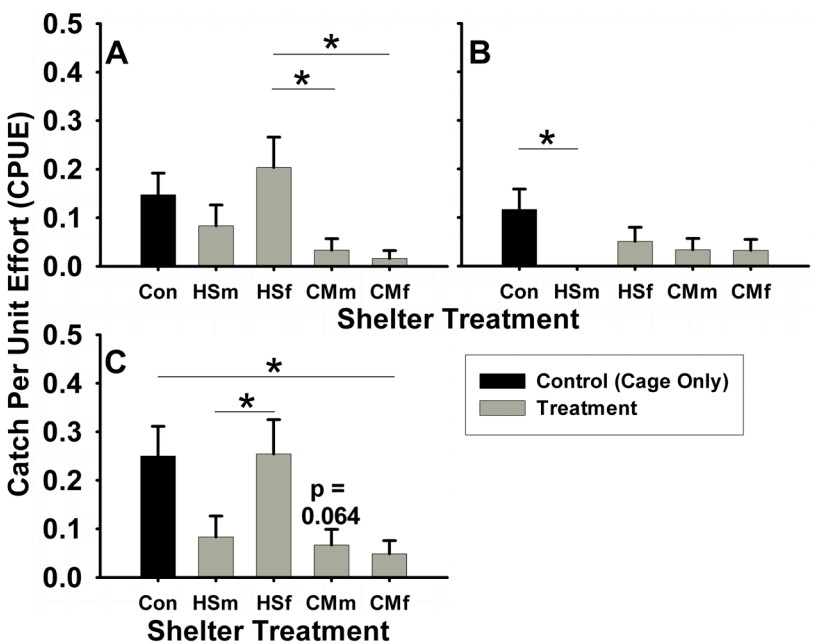

**Figure 4 Summer shelter choice in *Hemigrapsus sanguineus*.** Stated in catch per unit effort (CPUE, in number of animals caught per number of tubes (n = 60 tubes for each bar)) mean ± SE. Treatments were: HSm = *H. sanguineus* male, HSf = *H. sanguineus* female, CMm = *Carcinus maenas* male, CMf = *C. maenas* female. Results for shelter occupancy are shown for (A) intertidal male *H. sanguineus*, (B) intertidal female *H. sanguineus*, and (C) overall intertidal *H. sanguineus*. Significance between treatments indicated by brackets: "*" = p < 0.05, "**" = p < 0.01, "***" = p < 0.001.

subtidal and intertidal summer data are included together, these treatment differences still hold in *H. sanguineus* residency (two-way ANOVA, N = 600, Tidal Deployment: F = 32.83, 1 d.f., p < 0.0001, Treatment: F = 4.54, 4 d.f., p = 0.0012, Tidal Deployment*Treatment, F = 3.698, 4 d.f., p = 0.0055).

In laboratory direct interspecific and intraspecific individual competition, *H. sanguineus* is more likely to use shelter than *C. maenas* (Fisher's Exact Test, p = 0.0020, N = 64) (Fig. 5A). In interspecific competition experiments, *H. sanguineus* utilized ("won") shelter 32.8% of the time, whereas *C. maenas* only utilized shelter 9.1% of the time. While this comparison is interesting on its surface, intraspecific competition puts this in context. When *C. maenas* competes against an individual *C. maenas*, one of the crabs utilizes the shelter 20.1% of the time (N = 29), which is not significantly different than its 9.1% shelter from when it was competing against *H. sanguineus* (Fisher's Exact Test, p = 0.1818, N = 93). The same result is observed for *H. sanguineus*, which uses shelter at 33.3% rate (N = 39) when competing against another *H. sanguineus*, which is also not significantly different in its shelter use from when it was competing against *C. maenas* (Fisher's Exact Test, p = 0.3019, N = 103). When analyzed by the effect of competitor size (Fig. 5A), *H. sanguineus* had a significant linear regression for shelter use (y = −0.1835x + 0.5728, $R^2$ = 0.7449, p = 0.0027), but *C. maenas*' use of shelter was not significant by linear regression (p = 0.49). When crabs of the same approximate size are compared (1:1 ratio group, N = 17), *H. sanguineus* is significantly more likely to end

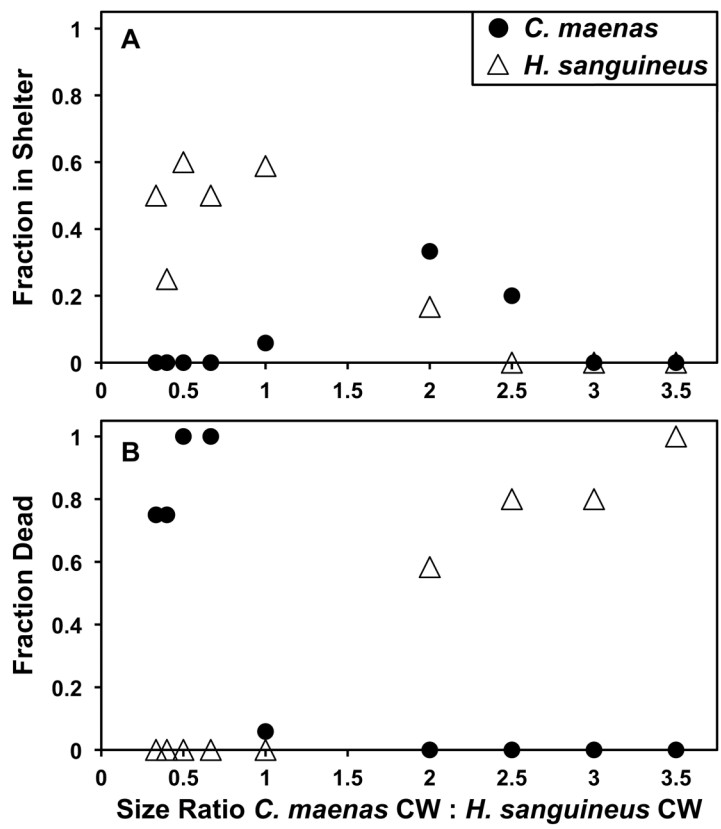

**Figure 5 Laboratory shelter competition between *Hemigrapsus sanguineus* and *Carcinus maenas.*** (A) Fraction of each species in shelter after 24 h of direct shelter competition between an individual *H. sanguineus* and *C. maenas*. At the left side of the x-axis, *H. sanguineus* are larger by carapace width (CW), and on the right, *C. maenas* are larger. N is between 4 and 17 for each point. (B) Fraction of each species dead after 24 h of direct shelter competition between an individual *H. sanguineus* and *C. maenas*. At the left side of the x-axis, *H. sanguineus* are larger, and on the right, *C. maenas* are larger. N is between 4 and 17 for each point.

up in the shelter than *C. maenas* in interspecific trials (Fisher's Exact Test, p = 0.0024, N = 17). However, neither is more likely to be injured or die when evenly matched by size. In interspecific trials, both *H. sanguineus* and *C. maenas* were killed more often when they were smaller than the opposing crab (Fig. 5B), and linear regressions explain a large amount of the variation of death rates in both species. Competitor size explains 96% of the variation in death rates of *H. sanguineus* (y = −0.3459x + −0.1805 ($R^2$ = 0.9607, p = 3.555 × $10^{-6}$)), and 67% of that in *C. maenas* death rates (y = −0.3101x + 0.8744 ($R^2$ = 0.6704, p = 0.0069)). *C. maenas* and *H. sanguineus* both exhibit cannibalism when large differences in carapace width exist (see Supplemental Data), and readily prey upon interspecific individuals.

## DISCUSSION

All collected species showed an increased use of shelter during warmer months, probably correlated with the increased predation in this system with increasing seasonal temperature (*Clark, 1968*; *O'Neill & Cobb, 1979*) and increased need for seasonal mating

and reproduction. Specifically, *Myoxocephalus aenaeus* and *Pagurus* spp. shelter competition with these decapods may have a greater impact during spring and summer months. Decapods *Carcinus maenas* and the *Pagurus* spp. complex showed peak shelter use in the spring deployments, and both of the fish species (*Gobiosoma ginsburgi* and *M. aenaeus*) peaked during summer. Other *Gobiosoma* spp. exhibit seasonal patterns in shelter use, primarily with an increase in use of seagrass shelter in July and October (*King & Sheridan, 2006*). Shelter competition could further pressure *G. ginsburgi* as *C. maenas* invades further into Canada in the North Atlantic (*Roman & Darling, 2007*), and this shelter competition will likely affect congener *Gobiosoma* species in areas other than the LIS at risk of *C. maenas* invasion.

Some species were collected in these shelters during this work, but not in sufficient numbers for seasonal or treatment analysis. The effects on the following species could not be assessed (see Supplemental Data): the oyster drill *Nucella lapillus* (6 collected), shrimp *Palaemontes* spp. (11 collected), *Crangon septemspinosa* (5 collected), *Cancer irroratus* (5 collected), *Panopeus herbstii* (21 collected), *Dyspanopeus sayi* (9 collected), Xanthid crab settlers (8 collected), *Pholis gunnelus* (1 collected), *Anguilla rostrata* (4 collected), *Littorina littorea* (17 collected), and the cunner *Tautogolabrus adspersus* (17 collected). These collected species could just be in lower numbers, not typically use shelter (not likely for the crab species), or be adversely affected by the presence of the two crabs. Determining the proximal cause requires further comparative work, but in the Gulf of Maine, both *Hemigrapsus sanguineus* and *C. maenas* make intertidal shelter maladaptive for native species (*Griffen & Riley, 2015*). If there is any intraguild predation interference on other community members as *Griffen & Byers (2006)* and *Griffen & Byers (2009)* found between *H. sanguineus* and *C. maenas*, this merits additional investigation with respect to shelter. It is also typically difficult to dislodge a resident decapod from a shelter, but some species are able to do this, such as the invasive *Eriocheir sinensis* (*Gilbey, Attrill & Coleman, 2008*), and in *H. sanguineus* and *Hemigrapsus oregonensis* (*Jensen, McDonald & Armstrong, 2002*; *Lohrer & Whitlatch, 2002*). If LIS shelter-using species cannot dislodge *H. sanguineus* or *C. maenas*, they may just avoid the shelter altogether, resulting in our undersampling of those species.

Species within the LIS community also responded to the treatments in our experiments. *Pagurus* spp. avoided *H. sanguineus* treatments, perhaps indicating previous individual experience with them as a predator. This was observed with induced responses to *C. maenas* predators in *Mytilus edulis* mussels (*Freeman & Byers, 2006*), and underscores how changes in dominance by each crab species could impact LIS community members, especially bivalve populations. *C. maenas* consistently consumes more blue mussels than *H. sanguineus* and per capita predation on mussels decreases with interspecific and intraspecific interference between and within *H. sanguineus* and *C. maenas* (*Griffen, 2006*). Prior to this work, *Pagurus* spp. was not specifically documented as potential prey of *H. sanguineus* (*Brousseau & Baglivo, 2005*), but crustacean parts were found in the stomachs of *H. sanguineus* (*Ledesma & O'Connor, 2001*).

Seaboard gobies, *G. ginsburgi*, actively avoided *C. maenas* treatments, indicating likely predation, but showed no aversion to *H. sanguineus* present in shelters, likely since

interactions with this species are unlikely in the subtidal. Congener *Gobiosoma* spp. in non-invaded areas would be expected to experience the same risk and respond in the same way. Newly-invaded communities are negatively impacted by *C. maenas* (*Walton et al., 2002*), and it likely had a similar impact upon its invasion to eastern North America (*Freeman et al., 2014*; *Griffen & Riley, 2015*). Although not much is known about *G. ginsburgi*'s specific behavior, congeners prefer woody debris as shelter (*Everett & Ruiz, 1993*), and males guard eggs even through near-lethal levels of hypoxia (*Breitburg, 1992*). There is high tidal flow at our study site, which are the preferred conditions for congener *Gobiosoma* spp. (*Tolley et al., 2006*), but there may also be a population that moves up the estuary (*Schultz et al., 2003*) that merits investigation. *Gobiosoma robustum* chooses sand over seagrass shelter in the presence of a predator (*Schofield, 2003*); therefore, a similar pattern in *G. ginsburgi* would mean a greater use of sand, and thus greater exposure to finfish predators in areas where *C. maenas* is present.

No treatment was preferred by the grubby *M. aenaeus*, but it is known to be a predator of small decapods (*van der Meeren, 2000*). The area where we deployed these shelters does not have much vegetation and therefore is not a likely nursery habitat for *M. aenaeus* (*Lazzari, Sherman & Kanwit, 2003*), but it did still show seasonal shelter use in this system. Use of these shelters may be due to the slower escape response accelerations and velocities found in this genus than other teleosts (*Jordan, Herbert & Steffensen, 2005*), along with their role as an active winter predator of small lobsters (*van der Meeren, 2000*). The lack of correlation of *M. aenaeus* shelter use with treatments in our experiments demonstrated no interaction with *C. maenas* and *H. sanguineus.*

In this part of the LIS, *C. maenas* utilized artificial shelters far more in the subtidal than the intertidal, which is a reflection of its higher subtidal abundance, but this exclusively subtidal-use phenomenon has not been observed in other *C. maenas*-invaded areas, nor in its native European range (*Amaral et al., 2009*). The dominance of competing *H. sanguineus* in the intertidal zone of the LIS likely decreases the availability of these shelters for native species as it does in the Gulf of Maine (*Griffen & Riley, 2015*). However, our field experiments do not provide evidence of a *H. sanguineus*-mediated decrease in the *C. maenas* utilization of artificial shelters in the intertidal nor subtidal, even though we saw a difference in sheltering patterns between these two in the laboratory when pitted against one another. Ultimately, there is no evidence from these data that *H. sanguineus* drive the difference in abundance of *C. maenas* between subtidal and intertidal zones. This is consistent with work in Dutch estuaries where both *H. sanguineus* and *H. takanoi* are invading and competing with *C. maenas* (*van den Brink, Wijnhoven & McLay, 2012*), but the decline in *C. maenas* seems to be due to overall decline in environmental quality. However, a difference could arise if *C. maenas* cannot readily use shelter as easily, and *H. sanguineus* was clearly the shelter winner in similar-sized crabs in our laboratory trials. This could feed into the probable negative impact *H. sanguineus* has on early settlement survival for this population of *C. maenas* crabs. Work done by *Lohrer & Whitlatch (2002)* corroborates this fact, as zero-year *C. maenas* were not found as often in field enclosures with *H. sanguineus*. From these data and the literature, we can postulate that adult *C. maenas* will shelter where it wants regardless of conspecific or heterospecific presence,

and the intertidal/subtidal pattern is likely due to interactions early in ontogeny (*Lohrer & Whitlatch, 2002*).

It is clear that both *C. maenas* and *H. sanguineus* avoided each another. While *C. maenas* does not avoid shelter regardless of presence of a competitor in summer data, it does show a trend for avoidance of *H. sanguineus* when all seasonal data is pooled. In fact, same-size *C. maenas* are inferior shelter competitors to *H. sanguineus* in laboratory experiments. One would expect a strong signal for avoidance of aggressive male crabs of conspecifics and heterospecifics to avoid injury, but only *H. sanguineus* showed an avoidance of conspecific males, and it also avoided shelters with either male or female *C. maenas* (Figs. 4C and S1). That *C. maenas* does not show stronger avoidance of the heterospecific *H. sanguineus* shelters of either sex makes them more likely to be negatively affected in the way we observed in the direct competition studies, whereas *H. sanguineus* seems to avoid this negative interaction.

When *H. sanguineus* is the same size as its competitor or larger, it is not only more likely to inhabit the shelter, but also more likely to kill its competitor, in this case *C. maenas*. *C. maenas* will do the same to a smaller *H. sanguineus*, but we saw that when competing against a conspecific *C. maenas* it is less likely to use shelter in the first place. *H. sanguineus* is likely to avoid potential competitors in the field, but these data are size-dependent, consistent with earlier work (*Lohrer & Whitlatch, 2002*). Thus, one can conclude that this inherent lower use of shelter by *C. maenas*, combined with the agonistic interaction with *H. sanguineus*, results in an increased exposure of *C. maenas* to predation and desiccation in the intertidal zone and partially results in *H. sanguineus* dominance of this zone. The limited use of the subtidal zone by *H. sanguineus* in seasonal and summer subtidal shelter deployments means that in this particular system, it does not often interact with *C. maenas* in the subtidal. Although *H. sanguineus* is sometimes found in large numbers subtidally, this has been in marinas not along rocky shorelines (*Gilman & Grace, 2009*). This adds a critical dynamic to interspecific interaction for *H. sanguineus* in the intertidal zone: as it establishes in a new area, it will consume all likely competitors, dominate shelter use, and result in greater exposure of competitors to both terrestrial and marine predation (at high tide).

Shelter use data for *H. sanguineus* shows avoidance of potentially dangerous heterospecifics (*C. maenas* of both sexes) and conspecific male shelters and a preference for conspecific female shelters and unoccupied control shelters. This may enhance *H. sanguineus*' survival and reproduction through avoidance of antagonistic interactions like predation and competition, and indicates it can detect and use conspecific and heterospecific presence as an adult in addition to as a megalopa (*Steinberg, Epifanio & Andon, 2007*). In the laboratory direct competition individual match-ups, adult male *H. sanguineus* were more likely to take over vacant shelter. It is common in lobsters (*O'Neill & Cobb, 1979*) and pea crabs (*Grove & Woodin, 1996*) to use the presence of conspecifics to determine its use of shelter, but this avoidance behavior runs counter to the communal denning seen in lobster species (*Zimmer-Faust & Spanier, 1987*), which is often due to ontogeny (*Childress & Herrnkind, 1996*) or reproduction (*Bushmann & Atema, 1997*). In the subtidal, other decapods seem to avoid *H. sanguineus* treatments over

the other three treatments, underscoring how the presence of this species deters other decapods. Adult olfactory cues were implicated in metamorphosis and settlement of *H. sanguineus* (*Steinberg, Epifanio & Andon, 2007*; *Rasch & O'Connor, 2012*) near conspecifics, and now we can conclude that adult presence also has an impact upon small-scale spatial patterns, by indicating evaluation and mediation of risk by the adult animal.

From our data and the literature, changes in the abundances of these two invaders will clearly result in changes to LIS community shelter use. *C. maenas* does not compete for shelter as well as *H. sanguineus*, and the effects that the presence of each crab has on other community members' use of shelter means there are different effects of each on those populations' risk of predation. As such, changes in abundances of each of these two species may have different shelter and seasonal effects on different LIS community members, especially as they impact potential survival of *G. ginsburgi* and the *Pagurus* spp. complex, not to mention the potentially negative effects of the presence of these *H. sanguineus* and *C. maenas* have on each other.

## ACKNOWLEDGEMENTS

We thank the Department of Physiology and Neurobiology and Department of Marine Sciences at the University of Connecticut, J.L. Renfro, R.W. Whitlatch, A. Moiseff, and K. Schwenk for input on experimental design, and A. Hudson (González), J.S. Cobb, M. Gilman, C. McGinnis, N. Hinds, I. Tornberg, and E. Limouze. L. Stefaniak provided valuable comments on the manuscript. This work was impossible without the help of numerous divers and boat operators, to whom the authors remain indebted.

### Funding

Funding of this work was provided by the Connecticut Sea Grant College Program (development funding); the Center for Environmental Sciences and Engineering at the University of Connecticut (semester of partial assistantship support); and the University of Connecticut's Outstanding Scholar Program (multi-year fellowship). The funders had no role in study design, data collection and analysis, decision to publish, or preparation of the manuscript.

### Competing Interests

The authors declare that they have no competing interests.

### Author Contributions

- David M. Hudson conceived and designed the experiments, performed the experiments, analyzed the data, contributed reagents/materials/analysis tools, wrote the paper, prepared figures and/or tables, reviewed drafts of the paper.
- Dugan Reagan performed the experiments, reviewed drafts of the paper.
- Joseph F. Crivello conceived and designed the experiments, contributed reagents/materials/analysis tools, reviewed drafts of the paper.

### Field Study Permissions

The following information was supplied relating to field study approvals (i.e., approving body and any reference numbers):

Connecticut Department of Environmental Protection, Scientific Collector's Permits # SC-06040 and SC-09015.

### Data Deposition

The raw data has been supplied as Supplemental Dataset Files.

### Supplemental Information

Supplemental information for this article can be found online at http://dx.doi.org/10.7717/peerj.2265#supplemental-information.

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
