# Peer review of "Community shelter use in response to two benthic decapod predators in the Long Island Sound"

_PeerJ, doi:10.7717/peerj.2265_

## Round 0.1 · original submission · Minor Revisions

Both reviewers have now submitted their report to me; while both were positive, they have also recommended some revisions. Most of the revisions are related to explanation of details. As well, one reviewer found the Introduction to be a bit short, while the other found the Results to be confusing. I suggest carefully considering their comments, and look forward to seeing an improved version of this manuscript.

·

Basic reporting

The article is generally very well written and shows some solid work and some thought through analysis. I have a few comments about the basic reporting style:

Introduction: The introduction seems to be a bit more difficult to read compared with the rest. It is almost as if the author was given a word limit and had to write as efficiently as possible. For example: “This requires animal evaluation of the cohabitation risk” This is so efficient in word count that it does not make for easy reading. Don’t be afraid of using conjunctions and making the sentences more relaxed.

Introduction: Why separate males and females? The hypothesis behind this idea is not explained

Line 217: rewrite sentence to be clearer. “in the summer intertidal experiments differences in male and female H. sanguineus behaviour was observed”.

Line 315: “….our study site, 'which are the' preferred conditions…..”

Throughout: Write species names in full at the start of the sentence

Line 323: is not 'a' likely nursery habitat

Line 324-326: “Use of these shelters may be due to the slower escape response accelerations and velocities found in this genus (Jordan, Herbert & Steffensen, 2005…” slower than what?

Lines 368-373: can you explain more about the effect of male or female treatments? E.g. C. maenas avoiding male H. sanguineus, but not females….why is that?

Line 638: “weeks, the diagrammed an artificial shelter” Either remove the ‘an’ or ‘the diagrammed’.

Figure 3. Figure labels are missing.

Line 664: Change ‘(upper graph)’ to ‘(top graph)’ and ‘(below graph)’ to ‘(lower grapsh)’ or label the graphs A and B.

Line 662-670: Missing a lot of full stops here, it make it a bit difficult to read.

Experimental design

A nice and concise experimental design. Obviously a lot of work went into this study!

It is not entirely clear, but I understand you used a single individual in the apparatus as 'bait'. Can you explain why one single individual would have had such an influence on the species that entered the tubes?

How were the tubes attached to each other? were there many species occupying the spaces outside of the tubes, and would this have an influence what what species initially approached the tubes?

Validity of the findings

The findings are valid and give another example of the community effects of an invasive crab.

Can you explain how the experimental set up relates specifically to the local environment? Does it imitate it?

Line 329-332: 'In this part of the LIS, C. maenas utilized artificial shelters far more in the subtidal than the intertidal, which is a reflection of its higher subtidal abundance, but this subtidal-use phenomenon has not been observed in other C. maenas-invaded areas, nor in its native European range'
In my experience, at least in Europe, adult C. maenas is found a lot more abundantly in the subtidal areas (particularly mussle beds) compared with the intertidal areas, especiallyin the winter because they migrate there for more food availability.

Additional comments

I enjoyed reading your manuscript. It is closely related to one that I am working on myself at the moment. Your experiment was extensive and generally well designed and your analysis and application is well thought out.
I had a few points to draw your attention to, Generally the writing style is good, but could be a bit more relaxed, and there were a few typos.

Reviewer 2 ·

Basic reporting

- L 61-63: there are more recent reports on the invasion of H. sanguine and H. taken on the european coast (Gothland et al, 2013; Gothland et al, 2014).
- Figure panels are not properly labeled (Figs 1, 3 & 4).
- I think that Figure caption should not give interpretation of the results (e.g. Fig 3: "shows...greater use of ...", "showed no significant"). Keep the caption as describing what the results are.
- the Results section is rather long (5 p) and I would encourage the authors to shorten this section to ease the reading

Experimental design

No comments

Validity of the findings

- Though the statistical analysis are (except for regression, see below) appropriate, the statistical tests used are not all mentioned in the Methods section (2 Way-ANOVA, Fischer's Exact test, Students' t-test are lacking).
- Fig. 3: it is very surprising that HSF treatment (on what should be panel C) is not significantly different from other treatments.
- I am not sure that Fig. 5 is needed
- Fig 7 includes regressions. These regressions are, to my opinion, useless, since they are not mentioned in the text, neither in the results, nor in the discussion. Furthermore, the regression for C. means is not significant (r=0.264, n=10) and should thus not be drawn.
L 251-252: It is surprising that 9.1% is not significantly different from 20.1%...

Additional comments

An interesting and generally well written paper. The results are a bit confusing because too long and the paper would gain in clarity if the authors could shorten the Results section.

---

## Round 0.2 · Minor Revisions

This version is much improved, but reviewer number 1 rightly still has some concerns about the statistics, and I believe their comments to be fair. Please add more information to help clarify the regression information as commented upon by the reviewer. Based on this, my decision is 'minor revisions' are still needed.

Reviewer 2 ·

Basic reporting

The manuscript has been greatly improved. However, I still have some comments about statistics.

Experimental design

OK

Validity of the findings

In their rebuttal, authors explained that "regressions were significant when done in R statistical software" and "all regressions had a p<0.01". This is still difficult to understand, since, when looking at statistical tables, a regression with r=0.264 and n=10 is not significant for alpha=0.05. Actually, a r=0.264 would be significant for n>50...(see e.g. tables in Zar 2010). Though this regression does not appear anymore in the ms (why...?) and other regressions are clearly significative, this plant seeds of doubt about the statistical analysis and I encourage the authors to check all their statistical results.
When I underlined that it was surprising that 9.1% was not significantly different from 20.1%, the authors replied "numbers are numbers". I agree, but statistics are statistics and a Student's t-test is not appropriate to compare frequencies! Furthermore, the authors add that this was maybe due to the "low catch rate": does it mean that "classical" parametrical statistics are not appropriate here and that non-parametric tests should be used?

Additional comments

The paper is very interesting and based an a well designed and original approach. Just clean up the statistics and it will be great!

---

## Round 0.3 · accepted · Accept

The final details regarding statistics have been adequately addressed, and this manuscript is now ready to be accepted. I look forward to seeing this article published.